

# 1 Magnetic dipolarizations inside geosynchronous orbit with tailward
# 2 ions flow

Xiaoying Sun[1,2], Weining William Liu [1], Suping Duan[1]
[1] State Key Laboratory of Space Weather, National Space Science Center (NSSC),Chinese Academy of Sciences (CAS),
Beijing, 100190, China
[2]University of Chinese Academy of Sciences, Beijing, 100049, China
*Correspondence to*: W .W. Liu (wliu@nssc.ac.cn), Suping Duan (spduan@nssc.ac.cn)
**Abstract.** Electromagnetic field and plasma data from the Time History of Events and Macroscale Interactions
duringSubstorms (THEMIS) near-Earth probes are used to investigate magnetic dipolarizations inside geosynchronous orbit
on 27 August 2014 during an intense substorm with $AE_{max}$ ~ 1000nT. THEMIS-D (TH-D) was located inside
geosynchronous orbit around midnight in the interval from 09:25 UT to 09:55 UT. During this period two distinct magnetic
dipolarizations with tailward ions flow are observed by TH-D. The first one is displayed by magnetic elevation angle
increase from 15 degree to 25 degree around 09:30:40UT. The tailward perpendicular velocity is $V_{\perp x}$ ~ -50$km/s$. The
second one is presented by the elevation angle increase from 25 degree to 45 degree around 09:36 UT. And the tailward
perpendicular velocity is $V_{\perp x}$ ~ -70 $km/s$. These two significant dipolarizations are accompanied with the sharp increase in
the energy flux of energetic electron inside geosynchronous. After 5 min expanding of near-Earth plasma sheet (NEPS),
THEMIS-E (TH-E) located outside geosynchronous orbit also detects this tailward expanding plasma sheet with ion flow -
150 $km/s$. The dipolarization propagates tailward with speed -47 $km/s$, along 2.2 $R_E$ distance in the X direction between
TH-D and TH-E within 5 min. These dipolarizations with tailward ions flow observed inside geosynchronous orbit indicate
new energy transfer path in the inner magnetosphere during substorms.
Keywords: Magnetic dipolarization, tailward ions flow, near-Earth plasma sheet, intense substorm

## 22 Introduction

Magnetic dipolarization can be observed at or inside geosynchronous orbit during intense substorms with high $AE$ index
($AE > 500$ nT) [e.g., Dai et al., 2015; Nagai, 1982; Nosé et al., 2014; Ohtani et al., 2018]. Dipolarizations are marked by the
magneticelevation angle increase with the decrease in the radial components of $B_x$ and $B_y$, and the increase in the $B_z$
component [Liu and Liang, 2009; Duan et al., 2011; Dai et al., 2014, 2015]. Ohtani et al. [2018] presented the statistics
characteristics of magnetic dipolarizations inside geosynchronous orbit. They reported that the dipolarization region
expanded in the azimuthal direction with speed 60 $km/s$ at 5.5 $R_E$. Using multiple satellites conjunction observations at or



inside geosynchronous orbit, Dai et al. [2015] reported thatthe large dipolarization electric field was associated with
substorm injection of MeV electrons into the inner magnetosphere (r < 6.6 $R_E$).

Magnetic dipolarizations are accompanied with complex ions bulk flow in the near-Earth plasma sheet (NEPS) [e.g., Duan et
al., 2008; Liang et al., 2009]. Especially, it is more complex in the inner edge of NESP. Usually, the substorm-associated
dipolarizations in the NEPS are accompanied with earthward ions bulk flow [e.g., Angelopoulos et al., 1992; Baumjohann et
al., 1999; Duan et al., 2011; Liang et al., 2008; Liu et al., 2008; Nakamura et al., 2009; Shiokawa et al., 1998]. According
conjunction observations of THEMIS multiple probes in the NEPS, Duan et al. [2011] pointed out that the dipolarization at
inner edge of the near-Earth plasma sheet had no one-to-one relationship with the earthward ions bulk flow. Lui et al. [1999]
pointed out that dipolarization at $X \sim 10\,R_E$ was detected with tailward flow. Inside geosynchronous orbit, magnetic
dipolarizations were detected with earthward ions bulk flow [Dai et al., 2015].

Near-Earth Dipolarizations with low frequency waves are detected with thermal ions and electron energization [e.g., Dai et
al., 2015; Liang et al., 2008, 2009; Nosé et al., 2014; Ohtani et al., 2018]. These energetic particles are main source of inner
magnetosphere during substorms and storms. Nosé et al. [2014] proposed that the dipolarizations associated with low
frequency fluctuations were observed in the inner magnetosphere during the storm main phase. These low frequency
electromagnetic waves can accelerate$O^+$ ions in the perpendicular direction. The low frequency waves can accelerate
particles crossing the magnetic field with large perpendicular electric field [e.g., Dai et al., 2014, 2015; Duan et al., 2016;
Nosé et al., 2014]. Usually, Dipolarization associated dispersionless energetic particle injections is accompanied with
earthward ions bulk flow in the NEPS [Dai et al., 2015]. But few reports show that dipolarizations with the sharply increase
in the energy flux of energetic particlesassociated with tailward ions flow at or inside geosynchronous orbit.

The ballooning mode occurred in the near-Earth plasma sheet are associated with tailward expansion of plasma sheet during
substorms [Liu, 1997; Liu et al., 2008; Liu and Liang, 2009; Liang et al., 2009; Saito et al., 2008]. Liu et al. [2008] pointed
out that the ballooning mode could excite a quasi-electrostatic field a few minutes before local current disruption and that the
perturbations associated with ballooning instability propagated downtail.

In this paper we present a dipolarization with tailward ions flow inside geosynchronous orbit during an intense substorm
expansion phase. The observations in detail of an intense substorm on 27 August 2014 by TH-D and TH-E are presented in
section 2. Discussions and conclusions of our observation results are displayed in the last section.



## Observations of an intense substorm on 27 August 2014

The OMNI data of the solar wind, interplanetary magnetic field (IMF) and geomagnetic field index $Dst$ and $AE$ during a storm on August 27, 2014 are presented in Figure 1. The minimum value of $Dst$ index is about -80 nT, as shown in Figure 1e, imply that a moderate storm take placed. During the main phase of this moderate storm, there is an intense substorm with the $AE$ maximum value 1273 nT. The beginning time of this intense substorm expansion phase is around 09:31 UT with decrease in the $AL$ index. A significant substorm enhancement occur around 09:48 UT with sharply decrease in the $AL$ index and increase in the $AE$ index.

During this intense substorm, THEMIS probes [Angelopoulos, 2008], such as TH-D and TH-E are both located in the near-Earth magnetotail. Figure 2 displays the orbits of TH-D and TH-E from 09:20 to 10:00 UT in the SM coordinate system. At 09:30 UT, locations of these two spacecraft, are (-6.01, -0.06, 1.12) $R_E$ for TH-D, (-8.10, -2.28, 1.92) $R_E$ for TH-E, respectively. TH-D orbit plot presents that it is located inside geosynchronous orbit at the beginning time of this intense substorm expansion phase. On the other hand, TH-E is located outside geosynchronous orbit. These two spacecraft present good conjunction observations during this intense substorm expansion phase. The instruments adopted in our investigations are the fluxgate magnetometer (FGM) [Auster et al., 2008], the electrostatic analyzer (ESA) [McFadden et al., 2008], the electric field instrument (EFI) [Bonnell et al., 2008] and the solid state telescope (SST) on board the THEMIS probes.

Figure 3 shows the plasma parameters and the electromagnetic field detected by TH-D mostly inside geosynchronous orbit at about midnight section. The solar magnetic (SM) coordinate system is adopted. From top to bottom, panels are the total magnetic field value, $B_t$ and the $B_x$ component, the $B_y$ and $B_z$ components, the magnetic field elevation angle defined by $\theta = \tan^{-1}(B_z/(B_x{}^2 + B_y{}^2)^{1/2})$, the ion and electron density, temperature, the plasma beta value $\beta$, $\beta = 2\mu_0 nT/B^2$, which determines the location of the satellite [Miyashita et al., 2000], three components of ions bulk flow velocity parallel (black line) and perpendicular (red line) to the magnetic field, $V_x$, $V_y$ and $V_z$, three components of the electric field, the $E_x$ (red), the $E_y$ (black) and the $E_z$ (blue), there components of convection electric field from $\mathbf{V} \times \mathbf{B}$, the $E_{cx}$ (red), the $E_{cy}$ (black) and the $E_{cz}$ (blue), respectively. Figure 3 displays the distinct fluctuations of the magnetic field and plasma density and velocity around 09:30 UT and 09:36 UT, respectively. The magnetic elevation angle has two step clear enhancements as displayed in Figure 3c. The first increase in elevation angle is from about 15 degree to 25 degree during the interval from 09:30:34 UT to 09:30:54 UT, which are marked by the left two vertical dashed lines in Figure 3. The total magnetic field value and the $B_x$ component both decrease. The $B_z$ component increase weakly from about 35 nT to 45 nT. The $B_y$ component has obvious fluctuations around 0 nT. These magnetic signatures indicate a magnetic dipolarization take places inside geosynchronous orbit around (-6.10, -0.06, 0.43) $R_E$. During this weak magnetic field dipolarization, the plasma beta value, $\beta$, increases from around 0.5 to 1.0. The ion and electron density and temperature both increase. Accompanied this dipolarization the tailward ions bulk flow, $V_{//x} \sim -100 km/s$ and the perpendicular component to the magnetic field in the X direction, $V_{\perp x} \sim -50\ km/s$

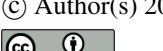



is detected by TH-D, as shown in Figure 3g. The perpendicular velocity in the Y direction is mainly dawnward at the
beginning time of this dipolarization, $V_{\perp y} \sim -30 km/s$. The electric field detected by TH-D also has large fluctuations with
negative $E_y$ value during the first depolarization as shown in Figure 3j. During the intervals from 09:30:34 UT to 09:30:54
UT the convection electric field direction is dawnward with large magnitude, $E_{cy} \sim -12\ mV/m$, as presented in Figure 3k.
The second magnetic field elevation angle increases sharply at around 09:36 as displayed in Figure 3c marked by the right
two vertical dashed lines. The elevation angle increases from about 25 degree to 45 degree during the interval from 09:36:06
UT to 09:36:21UT. The magnetic field has similar variations to the first dipolarization signatures. Especially, the second
dipolarization has larger elevation angle maximum value, ~ 45 degree, as marked by the fourth vertical dashed line in Figure
3c. During the second dipolarization the tailward ions bulk flow perpendicular to the magnetic field is also detected by TH-D,
$V_{\perp x} \sim -70\ km/s$, as presented in Figure 3g. Also the significant negative $E_y$ component is companied by this intense
dipolarization in Figure 3j and 3k.

Duringthe intervals of magnetic dipolarizations with tailward ions bulk flow detected by TH-D inside geosynchronous orbit,
TH-E observed very weak increase in the magnetic field elevation angle and the $B_z$ component around 09:35 UT and 09:41
UT, as shown in Figure 4b and 4c, about 5 min after two dipolarizations detected by TH-D. The ions and electrondensity
andtemperature increase weakly from very low value as displayed in Figure 4d and 4e. Outside geosynchronous orbit, TH-E
observed very low beta value, as shown in Figure 4f, $\beta \sim 0.01$ and $\beta \sim 0.2$ around 09:35 UT and 09:41 UT, respectively.
Interesting phenomena that the weak dipolarization was with the tailward ions bulk flow, $V_{//x} \sim -180 km/s$, is also detected
by TH-E around 09:35 UT as shown in Figure 4g. The perpendicular velocity is dominated in the negative Y direction, $V_{\perp y} \sim$
$-50\ km/s$.

Associated with the intense electric field observed by TH-D inside geosynchronous orbitduring this two dipolarizations, the
energy fluxes of energetic electrons, as shown in the second panel of Figure 5, with energy of 31 $keV$ (blue), 41 $keV$ (gray),
52 $keV$ (red), 65.5 $keV$ (black), 93 $keV$ (brown) and 139 $keV$ (purple) all simultaneously increase at 09:30:38 UT and
09:36:09UT detected by SST/TH-D, respectively. These energetic electrons have quasi-perpendicular pitch angle
distribution, as presented in the bottom panel of Figure 5.
**Discussion and conclusions**
The dipolarizations with tailward ions bulk flow inside geosynchronous orbit are investigated in our present paper.
Accompanied these dipolarizations the energy fluxes of energetic electrons with energy between 31 $keV$ and 139 $keV$
simultaneously increase inside geosynchronous orbit. According to these energetic electrons pitch angle distributions, it is
found that high energy electrons mainly in the quasi-perpendicular direction to the magnetic field, as shown in Figure 5. On





the other hand, the inductive electric field during these two magneticdipolarization is in the dawnward direction as display in
Figure 3j and 3k. Previous research work reported that the inductive electric field associated with substorm dipolarization
can accelerate particles in the near-Earth plasma sheet [Dai et al., 2014, 2015; Duan et al., 2016; Fu et al., 2011; Fok et al.,
2001; Liu et al., 2010; Lui et al., 1988, 1999; Nakamura et al., 2009; Nosé et al., 2014]. As shown in Figure 3j around
09:36:30 UT the inductive electric fields in the second dipolarization are dominated in the $E_y$ component with large negative
value, $E_y \sim -25mV/m$, and the X component also increase with negative value $E_x \sim -6mV/m$. This intense electric field can
drive ions moving into the tailward-dawnward direction. On the other hand, we can calculate the energy quantity relationship
between the electric field and energetic electrons. Estimating the energy of such intense $E_y$ in the distance of $\sim 1000\ km$ is
about $\sim 10^{-15}$ Joule. The energetic electrons with energy range from $31\ keV$ to $139\ keV$ are in the same energy order $\sim 10^{-15}$
$^{15}$Joule. It is inferred that the intense $E_y$ can perpendicularly accelerate electrons to tens $keV$ state.

Dipolarizations occurring at the inner edge of plasma sheet are complicated with disturbances of ions bulk flow and
electromagnetic field. Lui et al. [1999] pointed out that near-Earth dipolarization was a non-MHD process and was also
accompanied with tailward ions flow.Our observations of dipolarizations inside geosynchronous orbit are also associated
with tailward ions flow. This result is consistent with the report proposed by Liu et al. [2008] that the perturbations
associated with the ballooning mode in the near-Earth plasma sheet propagating tailward. Based on the statistic studies, Nosé
et al. [2016] proposed that the occurrence probability of the dipolarizations in the inner magnetosphere had a peak at 21:00-
00:00 MLT. Our observations show that two distinct dipolarizations with tailward flow inside geosynchronous orbit are
detected by TH-D around 00:02 MLT and 00:05 MLT, respectively.

According to the distance between TH-D and TH-E, (-2.23, -2.30, 0.56)$R_E$, and the delay time of dipolarization from inside
to outside geosynchronous orbit, $\sim$5 min, the dipolarization propagating speed or the plasma sheet expanding speed can be
estimated as $V_x \sim -47\ km/s, V_y \sim -48\ km/s, V_z \sim 12\ km/s$, respectively. Liou et al. [2002] proposed that the dipolarization
region expanding speed was $\sim 60\ km/s$ westward at geosynchronous. Comparing observations between TH-D and TH-E in
our investigations, the azimuth speed of dipolarization region is obtained $\sim 48\ km/s$. These two observational results are
consistent with each other.

Based on the above observation analysis, we can draw the results as following.  Two distinct magnetic dipolarizations with
tailward ions flow are observed by TH-D inside geosynchronous orbit on 27 August 2014 during the intense substorm with
$AE_{max} \sim 1000nT$. TH-D was located inside geosynchronous orbit around midnight in the interval from 09:20 UT to 10:00
UT. The first dipolarization is displayed by magnetic elevation angle increase from 15 degree to 25 degree around
09:30:40UT. The second one is presented by the elevation angle increase from 25 degree to 45 degree around 09:36 UT.
These two significant dipolarizations are accompanied with the energy flux of energetic electrons simultaneously increase

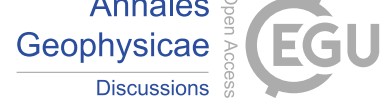

inside geosynchronous orbit. After 5 min expanding tailward of near-Earth plasma sheet, TH-E located outside geosynchronous orbit also detects this tailward expanding plasma sheet with ion flow -150 $km/s$. The dipolarization propagates tailward with speed -45 $km/s$, along 2 $R_E$ distant in the X direction between TH-D and TH-E within 5 min. These dipolarizations with tailward ion flow observed inside geosynchronous orbit indicate new energy transfer path in the inner magnetosphere during substorms.

**Acknowledgments**

We acknowledge NASA contract NAS5-02099 for use of data from the THEMIS Mission. Specifically: "D. Larson and R. P. Lin for use of SST data, C. W. Carlson and J. P. McFadden for use of ESA data; J. Bonnell and F. S. Mozer for use of the EFI data; K. H. Glassmeier, U. Auster and W. Baumjohann for the use of FGM data provided under the lead of the Technical University of Braunschweig and with financial support through the German Ministry for Economy and Technology and the German Center for Aviation and Space (DLR) under contract 50 OC 0302. The authors thank NASA CDAWeb and Taiwan AIDA for THEMIS data. The SYM − H index was provided by Data Analysis Center for Geomagnetism and Space Magnetism in Kyoto, Japan. This work is supported by the National Natural Science Foundation of China grants 41674167, 41731070 and 41574161; and in part by the Specialized Research Fund for State Key Laboratories.

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






Figure 1 The solar wind, IMF $B_z$ conditions and geomagnetic indices between 01:00 UT and 23:00 UT on August 27, 2014. From top to
bottom of (a ~ g) panels show the change of solar wind dynamic pressure (a), $B_{z,IMF}$ in GSM coordinate (b), the x component of the solar
wind flow speedin GSM coordinate (c), electric field$E$ (d), *AE/AU/AL* index (e),*SYM-H* indices (f), and *ASY-H* index (g). From left to right,
the verticaldotted lines in (a ~ g) panels marked the time 01:48 UT, 06:42 UT, 09:31 UT, 09:48 UT, 21:56 UT and 22:35 UT, respectively.

Figure 2 The orbits of TH-D and TH-E in the $X - Y_{SM}$ plane and the $X - Z_{SM}$ plane from 09:20 to 10:00 UT on 27 August 2014, which
were in the nightside magnetosphere. The arrow shows the flying direction of the satellites. TH-D is redand TH-E is blue.

Figure 3 The electromagnetic field and plasma parameters detected by TH-D in the intervals from 09:25 UT to 09:55 UT on August 27,
2014. The Solar Magnetic (SM)coordinated system is adopted. From top to bottom, panels showed that (a)the total magnetic field $B_t$
(black) and the X component $B_x$ (red), (b) the Y component $B_y$ (green) and the Z component $B_z$ (blue), (c)the magnetic field elevation
angle $\theta$; (d) ion and electron density $N_i$, $N_e$; (e)ion and electron temperature $T_i$, $T_e$; (f) plasma beta $\beta$; (g) the X component of ion
parallelvelocity and perpendicular velocity$V_{parx}$, $V_{perpx}$; (h) the Y component of ion parallel velocity and perpendicularvelocity
$V_{pary}$,$V_{perpy}$; (i) the Z component of ion parallel velocity and perpendicular velocity $V_{parz}$, $V_{perpz}$; (j) the electric field $E_x$ (red), $E_y$
(black), and $E_z$ (blue) by assuming $\mathbf{E} \cdot \mathbf{B} = 0$; (k) the electric field $E_{cx}$ (red), $E_{cy}$ (black), $E_{cz}$ (blue) calculated by $\mathbf{E} = \mathbf{B} \times \mathbf{V}$. The black
vertical dashed lines marked the time 09:30:34 UT,09:30:54UT, 09:36:06 UT and 09:36:21 UT, respectively.

Figure 4 The electromagnetic field and plasma parameters detected by TH-E in the intervals from 09:25 UT to 09:55 UT on August 27,
2014.The Figure format is the same as Figure 3.The black vertical dashed lines marked the time 09:35:36 UT and 09:36:18 UT.

Figure 5 The energy flux and pitch angle distribution of energetic electrons detected by SST/TH-D in 3 second time resolution. The red
vertical lines marked the time 09:30:38 UT and 09:36:09 UT.





Figure 1

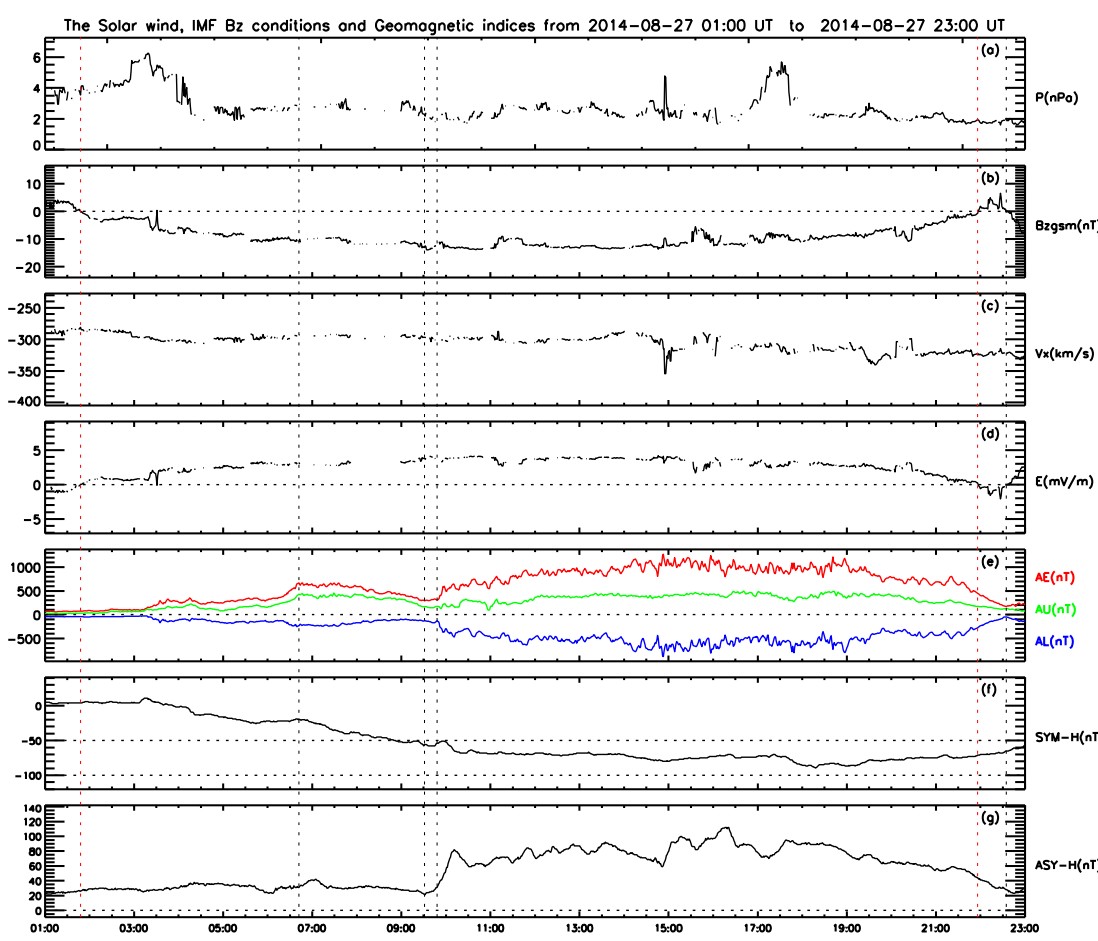

Figure 2

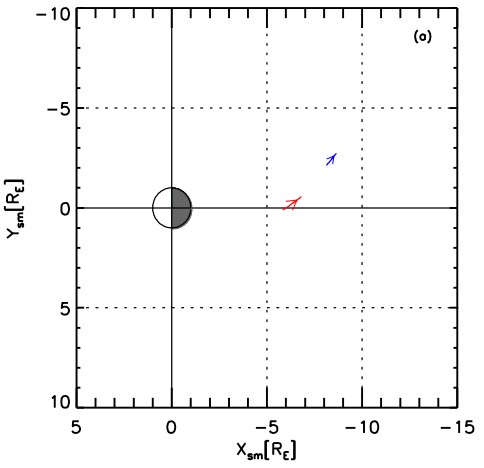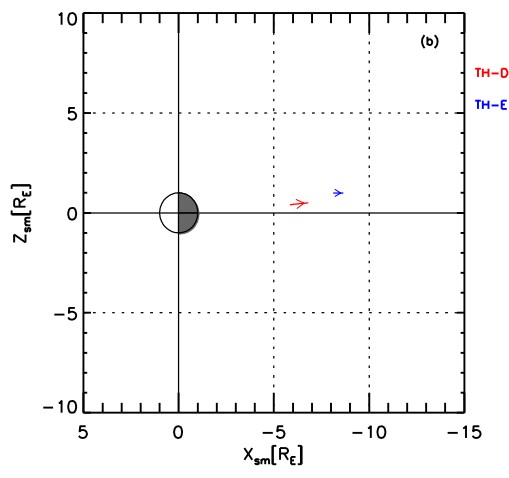



Figure 3








Figure 4






Figure 5

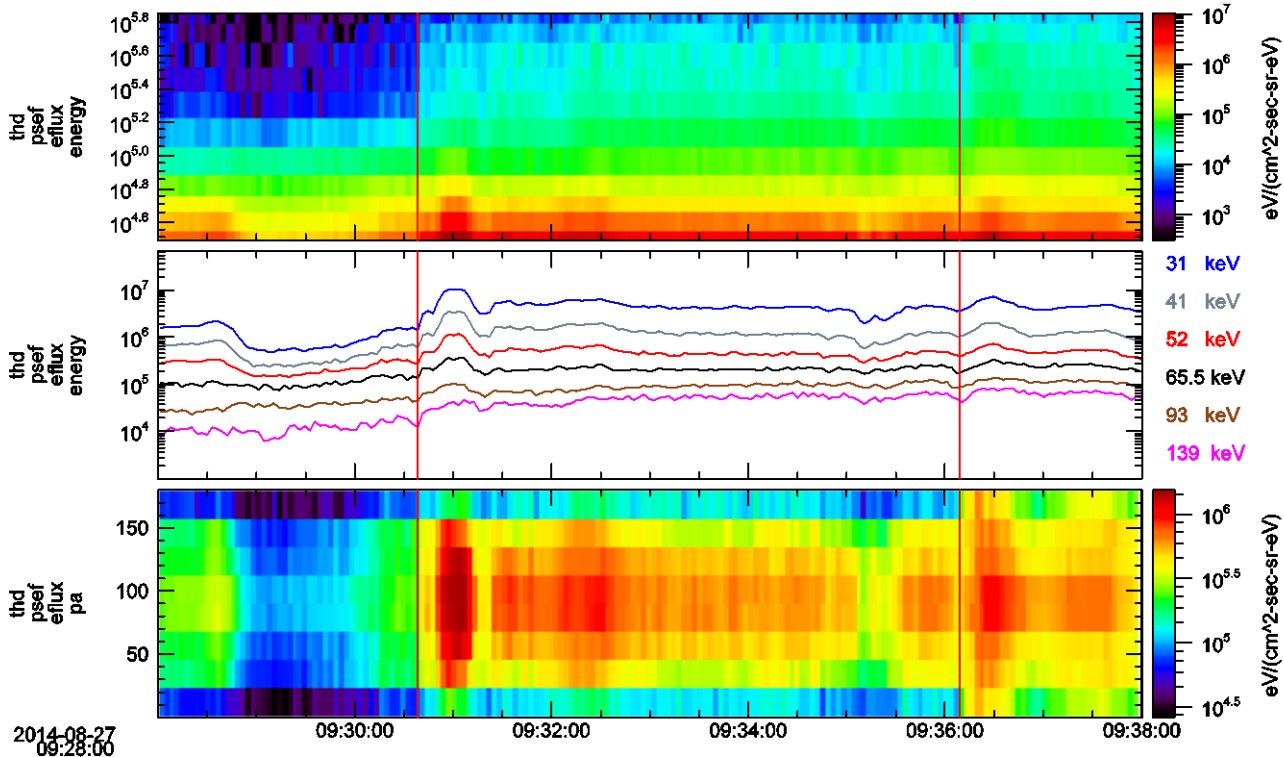

