# Peer review of "Magnetic dipolarizations inside geosynchronous orbit with tailward"

_Annales Geophysicae, 2018_

## Referee Comment (RC1) · Anonymous Referee #1 · 31 Dec 2018

The present paper studied two successive dipolarizations that were observed by the two THEMIS spacecraft located earthward and tailward of the geosynchronous orbit near midnight. These dipolarizations were accompanied by tailward flows. The authors concluded that the tailward flow propagates tailward in a speed of dipolarization region expansion, carrying energy. Before making decision for publication, however, I have a couple of major concerns which require additional data analysis and more detailed discussions.

The authors describe that THEMIS D observed the two successive dipolarizations at ∼0930 and ∼0936 UT, while THEMIS E observed only one dipolarization at ∼0936 UT. The authors associate the two dipolarizations with only one substorm that began at ∼0930 UT, and they link the dipolarization at THEMIS D at ∼0930 UT to the dipolarization at THEMIS E at ∼0936 UT that propagated tailward from the THEMIS D location at a speed of -47 km/s.

I, however, have a couple of concerns in the above interpretations. First, I am wondering whether the two successive dipolarizations are associated with a substorm or associated with a pseudosubstorm (pseudobreakup) and the following substorm. The authors state that THEMIS D observed the two dipolarizations, but THEMIS E observed only one dipolarization. Ohtani et al. (JGR, p. 19,355, 1993) showed that dipolarization associated with a pseudosubstorm is localized, while that associated with a substorm expands to a wide region. Hence there is a possibility that the ∼0930 UT dipolarization of the present event is localized at and near THEMIS D, associated with a pseudosubstorm, while the ∼0936 UT dipolarization expanded to both THEMIS D and E, associated with the following substorm. To verify the interpretation, the authors need to check ground substorm signatures, such as bay-type magnetic field changes, Pi2 and Pi1 pulsations, and auroral activity, at each ground station near the footprints of THEMIS D and E.

Second, I am wondering whether dipolarization at THEMIS D really occurred in two steps at ∼0930 and ∼0936 UT. In Figure 3, it seems that Bz continuously increased from ∼0930 or ∼0932 UT through ∼0937 UT and did not increase stepwise at ∼0936 UT. Furthermore, THEMIS E observed one dipolarization at ∼0937 UT. If dipolarization at THEMIS D occurred in two steps and if the dipolarization at THEMIS E is linked to the ∼0930 UT dipolarization at THEMIS D, how do the authors explain the lack of the second dipolarization at THEMIS E that could be linked to the ∼0936 UT dipolarization at THEMIS D? The ground signatures mentioned above may be helpful for this question.

After the additional analysis and discussions mentioned above, the dipolarizations at the two spacecraft can be linked, and hence the tailward propagation speed of the dipolarization region can be obtained in a more convincing way.
[Figure]

Other specific comments:

Lines 62-67: The maximum AE value of the substorm examined in the present study was ∼500 nT at ∼1010 UT, not 1273 nT at a later time. Hence this substorm should be moderate, not intense. After the present substorm, a lot of substorms or steady magnetospheric convection occurred during the storm main phase, and AE reached a peak of 1273 nT during one of these activities.

Line 90: The ion temperature was decreased, not increased, during the weak dipolarization at 0930 UT, while the ion density, the electron density, and the electron temperature were increased. This sentence is confusing, so please reword it.

Lines 108-109: It should be noted that these low beta values and its increase indicate that the spacecraft was in the lobe and moved to the plasma sheet boundary layer and then the plasma sheet. The parallel flow should have observed in the plasma sheet boundary layer.

Lines 128-129: The negative (tailward) Ex with the positive (northward) Bz corresponds to the duskward perpendicular flow, not the dawnward perpendicular flow. In the present event, the measured Ex is opposite to Ecx calculated from VxB. The measured electric field may need some caution, since it may include an offset and the contributions other than VxB.

Lines 143-148: In this paragraph, the authors discuss only the azimuthal speed of the dipolarization region expansion and do not discuss the tailward speed. Since the tailward speed is related to the main conclusion of the present study, it should be discussed as well.

Discussion: The current disruption model for substorm triggering proposed that current disruption and dipolarization launches a tailward propagating rarefaction wave, which should be accompanied by a fast earthward flow (e.g., Lui, JGR, p. 1849, 1991; Chao et al., PSS, p. 703, 1977). This is possibly in contrast to the present results. Hence

it might be good to discuss this discrepancy or how different the rarefaction wave proposed by the current disruption model and the tailward propagation of the tailward flow and dipolarization region discussed in the present paper.

Minor corrections:

Line 33: NESP –> NEPS

Line 35: Liang et al., 2008 –> 2009 ?

Line 42: Liang et al. (2008) should be deleted here because Liang et al. (2008) did not show magnetotail observations.

Lines 60-61: Dst –> Sym-H

Line 61: Figure 1e –> Figure 1f

There is no space between words in many places throughout the text. Put space between the words throughout the text.

---

## Editor Comment (EC1) · Anna Milillo (Editor) · 6 Feb 2019

This paper reports the observations of two dipolarizations linked to a substorm registered by the two THEMIS spacecraft E and D located one inside the geosynchronous orbit and the other tailward. The paper is well written. Essentially I agree with the other referee that some more check should be done to prove the double dopolarization occurrence. Also there is some confusion with the Electric field directions and flow velocity directions. I will recommend it for pubblication after these revisions. Minor comments line 33 NESP should be NEPS line 69: the z coordinates of the two s/c here are probably wrong, since both are in the plasma sheet , in fact in the figures 3 and 4 there are different values. there are many typos and missing spaces within the manuscript.

---

## Author Comment (AC1) · 20 Mar 2019

Comments: The present paper studied two successive dipolarizations that were observed by the two THEMIS spacecraft located earthward and tailward of the geosynchronous orbit near midnight. These dipolarizations were accompanied by tailward flows. The authors concluded that the tailward flow propagates tailward in a speed of dipolarization region expansion, carrying energy. Before making decision for publication, however, I have a couple of major concerns which require additional data analysis and more detailed discussions.

Responses: We thank you for your comments that help improving the manuscript. In light of your comments, we have revised the manuscript accordingly. Now one-to-one

responses are the following.

Comments: The authors describe that THEMIS D observed the two successive dipolarizations at _0930 and _0936 UT, while THEMIS E observed only one dipolarization at _0936 UT. The authors associate the two dipolarizations with only one substorm that began at _0930 UT, and they link the dipolarization at THEMIS D at _0930 UT to the dipolarization at THEMIS E at 0936 UT that propagated tailward from the THEMIS D location at a speed of - 47 km/s. I, however, have a couple of concerns in the above interpretations. First, I am wondering whether the two successive dipolarizations are associated with a substorm or associated with a pseudosubstorm (pseudobreakup) and the following substorm. The authors state that THEMIS D observed the two dipolarizations, but THEMIS E observed only one dipolarization. Ohtani et al. (JGR, p. 19,355, 1993) showed that dipolarization associated with a pseudosubstorm is localized, while that associated with a substorm expands to a wide region. Hence there is a possibility that the _0930 UT dipolarization of the present event is localized at and near THEMIS D, associated with a pseudosubstorm, while the _0936 UT dipolarization expanded to both THEMIS D and E, associated with the following substorm. To verify the interpretation, the authors need to check ground substorm signatures, such as bay-type magnetic field changes, Pi2 and Pi1 pulsations, and auroral activity, at each ground station near the footprints of THEMIS D and E.

Responses: Thank you for this comment. Firstly, there are multiple dipolarizations during a substorm as reported in Paper of Duan et al. 2011 AG (Duan, S. P., Liu, Z. X., Liang, J., Zhang, Y. C., and Chen, T.: Multiple magnetic dipolarizations observed by THEMIS during a substorm, Annales Geophysicae, 29, 331-339, 2011). The dipolarization at substorm onset is localized with small scale but at substorm enhancement during substorm expansion phase has large spatial scale.

Basing on your suggestions we have checked the ground magnetic field data and present the figures as following. Under the mapping of T96, at 09:30 UT, the footprint of TH-D was near the ground stations of WHIT (White Horse), FSIM (Fort Simpson),

ATHA (Athabasca), FSMI (Fort Smith) and LARG (La Ronge); the footprint of TH-E was near the ground stations of ATHA, FSMI, FSIM and LARG, which are shown in Figure 1'. Figure 2' and Figure 3' provide that the ground magnetic field signatures mark this substorm process as the two dashed vertical lines. The ground stations near the footprints of TH-D and E are listed in Table 1 as following.

The 09:30 UT dipolarization is associated with the substorm onset time as marked by the AL index and other ground substorm signatures, such as the bay disturbance and Pi2 plusations as shown in Figure 2' and Figure 3'. It is not associated with the Pseudosubstorm. This substorm dipolarization is accompanied by the plasma sheet expanding during the substorm expansion phase and propagates toward the magnetotail accompanied by the magnetic field fluctuations with tailward ions bulk flow.

On the other hand, at 09:30 UT TH-E is located in the outer magnetosphere, such as in the lobe, the plasma beta and number density are very low. Thus the location of TH-E is far away from the substorm onset region and it cannot detect substorm signatures, such as the magnetic dipolarization.

Ohtani et al. [1993, JGR] reported that the two successive pseudosubstorm during very weak AE index (<100nT) as shown in Figure 2 in their paper. This weak geomagnetic activity was possiblely associated with pseudosubstorm. But the geomagnetic activity in our research work is very intense during a moderate storm with AE index being very high ∼500nT during our two successive dipolarization. This is a signature of substorms.

At 09:36 UT TH-E is located at the plasma sheet boundary layer. The plasma density and temperature are both increasing, the plasma beta value also increase. These parameters indicated that the near-Earth plasma sheet swept over TH-E spacecraft. This magnetic field elevation angle increases mark near-Earth plasma sheet expansion from the substorm onset location. Thus the dipolarization detected by TH-E at 09:36 UT is associated with the 09:30 UT dipolarization observed by TH-D.

Comments: Second, I am wondering whether dipolarization at THEMIS D really occurred in two steps at _0930 and _0936 UT. In Figure 3, it seems that Bz continuously increased from _0930 or _0932 UT through _0937 UT and did not increase stepwise at _0936 UT. Furthermore, THEMIS E observed one dipolarization at _0937 UT. If dipolarization at THEMIS D occurred in two steps and if the dipolarization at THEMIS E is linked to the _0930 UT dipolarization at THEMIS D, how do the authors explain the lack of the second dipolarization at THEMIS E that could be linked to the _0936 UT dipolarization at THEMIS D? The ground signatures mentioned above may be helpful for this question.

Responses: Thank you for this comment. Yes, the Bz component continuously increased from 09:30 UT through 09:37 UT. But it has a sharp increase at 09:36 UT. On the other hand the magnetic field elevation angle as shown in Figure 3c also increased sharply at 09:36 UT. The second dipolarization observed by TH-D at 09:36 UT was also detected by TH-E at 09:41 UT as marked by the third dashed vertical line. Furthermore the energetic electron dispersionless injection, as shown in Figure 5, at 09:30 UT and 09:36UT also supported these dipolarizations inside the geo-synchrounous orbit. Yes, the ground magnetic field station data as shown above also provide the evidences of these two dipolarizations as shown in Figure 2' and Figure 3'.

Comments: After the additional analysis and discussions mentioned above, the dipolarizations at the two spacecraft can be linked, and hence the tailward propagation speed of the dipolarization region can be obtained in a more convincing way.

Responses: Thank you for this comment.

Other specific comments: Lines 62-67: The maximum AE value of the substorm examined in the present study was _500 nT at _1010 UT, not 1273 nT at a later time. Hence this substorm should be moderate, not intense. After the present substorm, a lot of substorms or steady magnetospheric convection occurred during the storm main phase, and AE reached a peak of 1273 nT during one of these activities.

Responses: Thank you for this comment. We have revised the data in our paper as 'During the main phase of this moderate storm, there is an intense substorm with the AE maximum value 1273 700 nT around 10:10 UT'.

Line 90: The ion temperature was decreased, not increased, during the weak dipolarization at 0930 UT, while the ion density, the electron density, and the electron temperature were increased. This sentence is confusing, so please reword it.

Responses: Thank you for this comment. We have revised this sentense in our paper as 'The electron density and temperature both increase. The ion density also increases. But ion temperature decreases' in the line 91-92.

Lines 108-109: It should be noted that these low beta values and its increase indicate that the spacecraft was in the lobe and moved to the plasma sheet boundary layer and then the plasma sheet. The parallel flow should have observed in the plasma sheet boundary layer.

Responses: Thank you for this comment. Yes, the parallel flow has observed in the plasma sheet boundary layer which has been mentioned in our paper, lines 109-100: '...the weak dipolarization was with the tailward ions bulk flow, V_//x ∼ -180 km/s, is also detected by TH-E around 09:35 UT as shown in Figure 4g'.

Lines 128-129: The negative (tailward) Ex with the positive (northward) Bz corresponds to the duskward perpendicular flow, not the dawnward perpendicular flow. In the present event, the measured Ex is opposite to Ecx calculated from VxB. The measured electric field may need some caution, since it may include an offset and the contributions other than VxB.

Responses: Thank you for this comment. Firstly, TH-D is located inside geosynchronous orbit. So the electric field is not dominated by the convection electric field calculated from VxB. Second, during substorm dipolarization the inductive electric field is significant as shown in Figure 3j. Thus, the detected electric field is different from
the convection electric field Ec as shown in Figure 3k.

Lines 143-148: In this paragraph, the authors discuss only the azimuthal speed of the dipolarization region expansion and do not discuss the tailward speed. Since the tailward speed is related to the main conclusion of the present study, it should be discussed as well.

Responses: Thank you for this comment. We have added the discussion of the tailward speed of the dipolarization in our paper line 149 to 153 as 'The dipolarization associated with the current disruption propagated tailward with speed Vx $\sim$ -100 km/s detected by THEMIS satellites in the near-Earth plasma sheet X$\sim$ -11RE [Liu et al., 2008]. It is larger than the dipolarization propagating speed from inside to outside geosynchronous orbit V_x $\sim$ -47 km/s . This different speeds of dipolarization propagating tailward imply that the magnitude of the dipolarization speed may be associated with its beginning location in magnetotail plasma sheet.'

Discussion: The current disruption model for substorm triggering proposed that current disruption and dipolarization launches a tailward propagating rarefaction wave, which should be accompanied by a fast earthward flow (e.g., Lui, JGR, p. 1849, 1991; Chao et al., PSS, p. 703, 1977). This is possibly in contrast to the present results. Hence it might be good to discuss this discrepancy or how different the rarefaction wave proposed by the current disruption model and the tailward propagation of the tailward flow and dipolarization region discussed in the present paper.

Responses: Thank you for this comment. The recommended references above have been cited in our paper as in line 155 to 158 'On the other hand, Lui [1991] reported that substorm disturbance propagated tailward through a rarefaction wave front accompanied by earthward flow during substorm expansion phase early period. Chao et al. [1977] proposed that the rarefaction wave propagating tailward was accompanied by the thinning of plasma sheet and earthward plasma flow. This earthward flow is possibly convection flow or outflow flow of magnetic reconnection from the middle

magnetotail.'

Minor corrections: Line 33: NESP –> NEPS

Line 35: Liang et al., 2008 –> 2009 ?

Line 42: Liang et al. (2008) should be deleted here because Liang et al. (2008) did not show magnetotail observations.

Lines 60-61: Dst –> Sym-H

Line 61: Figure 1e –> Figure 1f

Responses: Thank you for these comments. We have revised above words one-by-one with blue color characters in our paper.

There is no space between words in many places throughout the text. Put space between the words throughout the text.

Responses: Thank you for this comment. We have checked space between the words throughout the text.

Please also note the supplement to this comment:
https://www.ann-geophys-discuss.net/angeo-2018-128/angeo-2018-128-AC1-supplement.zip

Table 1   The geographic longitude, geographic latitude, geomagnetic longitude and geomagnetic latitude of three geomagnetic observatories and satellites, and the local time of these stations at 09:30 UT.

| Observatory or satellite | Geographic latitude (°) | Geographic longitude(°) | Geomagnetic latitude(°) | Geomagnetic longitude(°) | 09:30 UT ~ LT |
|---|---|---|---|---|---|
| TH-D | 55.8 | 233.6 | 60.4 | 292.5 | 01:04 |
| TH-E | 55.7 | 246.4 | 62.2 | 307.1 | 01:57 |
| FSIM | 61.8 | 238.8 | 65.7 | 184.6 | 01:25 |
| FSMI | 60.0 | 248.2 | 62.4 | 193.0 | 02:03 |
| WHIT | 61.0 | 224.8 | 64.0 | 279.5 | 00:29 |
| LARG | 55.2 | 254.7 | 62.8 | 317.3 | 02:29 |
| ATHA | 54.7 | 246.7 | 57.6 | 188.1 | 01:57 |

Spacecraft Footprints and Ground-Based Instruments

Northern Hemisphere   2014-08-27 08:00-10:00 UT

[Figure]

Figure 1'   The spacecraft footprints and Ground-Based Observatory.

**Fig. 1.**

[Figure]

Figure 2´   Geomagnetic field observed by FSIM, FSMI,WHIT, LARG and ATHA between 09:25 UT
and 09:55 UT.

**Fig. 2.**

[Figure]

Figure 3′   The Pi 2 observed by FSIM, FSMI, WHIT, LARG and ATHA between 09:25 and 09:55 UT.

**Fig. 3.**

---

## Author Comment (AC2) · 20 Mar 2019

Comments: This paper reports the observations of two dipolarizations linked to a substorm registered by the two THEMIS spacecraft E and D located one inside the geosynchronous orbit and the other tailward. The paper is well written. Essentially I agree with the other referee that some more check should be done to prove the double dipolarization occurrence. Also there is some confusion with the Electric field directions and flow velocity directions. I will recommend it for publication after these revisions.

Responses: We thank you for your comments that help improving the manuscript. In light of your comments, we have revised the manuscript accordingly.

Comments: Minor comments line 33 NESP should be NEPS line 69: the z coordinates

of the two s/c here are probably wrong, since both are in the plasma sheet, in fact in the figures 3 and 4 there are different values. there are many typos and missing spaces within the manuscript.

Responses: Thank you for these comments. According to your suggestion we have revised the spacecraft orbit data shown line 69 as "locations of these two spacecraft in SM coordinates are (-6.10, -0.06, 0.43) R_E for TH-D, and (-8.26, -2.28, 0.99) R_E for TH-E, respectively". During this intense geomagnetic activity, the magnetic equator plane tilt towards southward, the small Z coordinate of TH-E does not mean it is located in the plasma sheet based on the plasma density, temperature and beta value as in Figure 4 in our paper. We have checked space between the words throughout the text.